# Effects of Feeding Management System on Milk Production and Milk Quality from Sheep of the Turcana Breed

**DOI:** 10.3390/ani13182977

**Published:** 2023-09-20

**Authors:** Ion Valeriu Caraba, Marioara Nicoleta Caraba

**Affiliations:** 1Faculty of Bioengineering of Animal Resources, University of Life Sciences “King Mihai I” from Timisoara, Calea Aradului 119, RO 300645 Timisoara, Romania; caraba_i@animalsci-tm.ro; 2Department Biology-Chemistry, Faculty of Chemistry-Biology-Geography, West University of Timisoara, Pestalozzi 16, RO 300315 Timisoara, Romania

**Keywords:** milk, sheep, nutrition, quality, microorganisms, heavy metals

## Abstract

**Simple Summary:**

The identification of an efficient feeding system for sheep that determines, in the short period of lactation, an increase in the quantity and quality of milk represents a concern at the level of microfarms. The Turcana breed of sheep is a mixed breed, where the milk production is low compared to the specialized breeds for milk production. At the level of Romania, there is an absence of concrete studies regarding the feeding system and analysis of the quality of sheep’s milk. The purpose of our study was to monitor the effect of the administration of a fodder complex prepared in a microfarm to determine the increase in milk production and quality. The milk was analyzed on the basis of physical–biological parameters, the content of trace elements and heavy metals, and microbiological parameters. The values of the physicochemical parameters analyzed in the milk from the sheep with additional feed indicated an improvement in the quality of the milk, and the values of the microbiological parameters indicated a good state of health of the sheep. On the basis of the analysis of the content of heavy metals in milk, it was established that there are no sources of pollution in the grazing area.

**Abstract:**

Milk and dairy products are among the foods preferred by consumers, as they are rich in nutrients, have high biological values, are easily accessible, and present a low risk to health. This study aimed to comparatively analyze the milk from sheep of the Turcana breed that were subjected to different feeding systems. The milk from the sheep was analyzed quantitatively and qualitatively; in this sense, the following were determined: daily milk production (DMY), physical parameters (pH, freezing point), chemical composition (lactose (L), fats (F), total proteins (TP), non-fat solids (Snf)), the content of heavy metals and trace elements (Zn, Cd, Cu, Fe, Mn, Pb), and microbiological parameters (the number of somatic cells (SCC), the total number of aerobic mesophilic germs that develop at 30 °C (NTG), the number of coliform bacteria (CT), the number of *Staphylococcus aureus*). Administration of the fodder complex produced, on the farm determined a slight quantitative increase in milk production, as well as in fat, protein and lactose content. The content of trace elements Zn, Fe, and Mn registered increases in milk samples from sheep that were administered the fodder complex. The content of heavy metals did not indicate any source of pollution in the grazing area. Furthermore, the microbiological parameters were within the allowed limits, indicating a good general state of health at the emergency level and the absence of microbiological contamination of the milk samples.

## 1. Introduction

Sheep’s milk and dairy products derived from it have been an integral part of human nutrition for several centuries; their use in human nutrition is mainly due to the essential constituents: micro- and macronutrients (fatty acids, vitamins, trace elements, etc.) [1].

Knowledge of the chemical composition, physicochemical properties, microbial load, and nutritional value of sheep’s milk is essential because sheep’s milk is used to obtain dairy products. Consumers are showing an increasing interest in the nutritional and health aspects of sheep dairy products [2]. Sheep milk quality is influenced by several factors such as genetics (breed, genotype), animal health, physiological aspects (age, parity, stage of lactation), environment (season), and management (type of feed, rearing system, milking techniques), as well as their interactions [3,4,5].

Representative studies have been carried out on sheep’s milk to evaluate chemical composition, physicochemical properties, and microbiological composition [1,6,7,8], as well as the content of trace elements and heavy metals [9,10,11,12]. Moreover, studies have been carried out to capture the effects of other factors on the quantitative and qualitative production of sheep milk, e.g., the effects of the lactation stage and the reproduction period [13,14,15], and the influence of exploitation management at the level of farms and micro-farms, namely, the growing and feeding systems, which can improve milk quality [5,12,16].

Finding an efficient feeding method that ensures the improvement of milk quality by increasing the composition of fatty acids and the nutritional properties of sheep dairy products is a topic that has been addressed by several groups of researchers. Feeding strategies based on grazing combined with the administration of fodder complexes established in optimal proportions seem to be a solution to this situation. As a result, it is believed that an adequate and balanced consumption of fats is necessary; therefore, fatty acids should be part of the diet, but in an adequate ratio to provide the body with the necessary content. The results of the studies that followed the increase in the quantity and quality of sheep’s milk through the administration of additional fodder complexes were encouraging [17,18,19].

The breeding of sheep in Romania targets native breeds (Tigaie, Turcana) to ensure the production of lambs, as well as the production of milk. In small farms, this mixed production (meat, milk) is problematic from an economic point of view due to a reduced opportunity to sell lamb meat (relatively limited period of time, very small number of importers), the small amount of milk obtained after weaning the lambs up until the induction of estrus, and lower subsidies for raising sheep. Increasing the economic efficiency at the level of small farms can be achieved by increasing the production of lamb meat (by applying biotechnical methods of assisted reproduction for sheep, which can monitor the induction and synchronization of estrus in the off-season), increasing milk production and its quality in terms of fat and protein content (administration of additional fodder complexes in the period after the lambs are weaned), and increasing consumer interest in sheep’s milk products.

In general, sheep with mixed exploitation (meat/milk) have a quantitatively lower milk production compared to those specialized in milk production, since the duration of lactation is comparatively shorter. However, the milk has comparable protein and fat content to that from predominantly milk-producing sheep. Studies have been carried out on the evaluation of milk production in non-dairy sheep, but they were limited to evaluations of the chemical composition and the microbiological and physical properties of milk [8,19,20,21]. In Romania, the evaluation of milk quality in sheep breeds has been limited to analyses of the chemical composition of the milk [13,22].

The aim of our study was to compare the quantitative and qualitative parameters of milk from sheep of the Turcana breed reared in a classic feeding system (grazing) compared to those that were additionally given complex fodder. The Turcana breed is a breed with mixed exploitation; as a result, monitoring the effect of the administration of a fodder complex prepared in the microfarm with the aim of increasing milk production and milk quality represents the novelty of this study. In addition, through an analysis of the content of heavy metals and trace elements, an assessment of the quality of the environment in the grazing area was made possible.

## 2. Materials and Methods

### 2.1. Location of the Study

The milk under analysis came from a private farm of Turcana sheep located in the western region of Romania (latitude = 44°57′33.3″ N, longitude = 22°16′28.1″ E), at an altitude of approximately 395 m. The milk was collected over a period of 3 months: May, June, and July 2023.

### 2.2. Animal Feeding and Management

In this study, 200 Turcana sheep aged between 2 and 6 years were selected from a herd of 1000, which calved in the same month. The selection of sheep in the 2 experimental groups was made from the entire herd of the farm after separating the lambs and after delivery of the lambs for slaughter:-Group A—100 sheep that grazed freely on the pasture area for about 10 h;-Group B—100 sheep that grazed freely on the pasture area for about 10 h, but also received additional feed.

The additional feeding consisted of the administration of concentrated feed prepared on the farm in the amount of 2.00 kg/sheep. The concentrated feed had a composition of 45% maize, 20% barley, and 35% ProteMix (HL-TopMix, Sliven, Bulgaria). ProteMix contains the following analytical components, according to the technical data sheet of the product: crude protein 33%; crude cellulose 11%; crude fat 2.5%; crude ash 9%; calcium 1.3%; phosphorus 0.92%; UFC 1.02. It is recommended to be administered to sheep in a percentage of 35% in combination with barley and maize.

The present vegetation was represented by species of grassy plants characteristic of meadows and pastures. The grazing areas were divided equally into 5 plots of 2 ha each, separated by an electric fence. The animals from each experimental lot (n = 100) were left to graze in a sector of 2 ha. At intervals of 3 weeks, the sheep from each experimental batch were moved to a new grazing sector; thus, each experimental batch had a 2-ha grazing sector at its disposal.

Milk samples were collected by manual milking performed in the evening. The milk samples subjected to analysis were collected weekly; in total, 12 collection campaigns were carried out. To determine the amount of milk, the individual collection of each milk sample was carried out. Later, the milk from the sheep in the same group was collected together. After homogenization, these collections constituted the total milk sample from which 3 sub-samples were taken (to perform triplicate analysis of the milk) in sterile containers according to the type of analysis to be carried out: physical parameters (pH was determined immediately), chemical composition (lactose, fat, protein, non-fat solids (Snf), freezing point), content of heavy metals and trace elements, and microbiological analyses. The milk was stored and transported at a temperature of +4 °C, approximately 4 h before the time of the analyses.

The experimental procedures and animal care conditions followed the recommendation of European Union Directive 86/609/CEE and were approved by the committee of experts for ensuring the welfare of experimental animals from the University of Life Sciences “King Mihai I” from Timisoara.

### 2.3. Determination of Physicochemical Parameters

The physical parameters were represented by the pH value and the temperature. They were measured using a portable multimeter Multi 340i/SET WTW (Weilheim, Germany) equipped with a specific sensor for each parameter. Determination of pH was performed immediately after milking, and calibration was conducted with standard buffer solutions at pH 4.0 and 7.0 according to the manufacturer’s instructions.

The chemical composition of milk was determined by infrared analysis (FTIR interferometer—Fourier-transform infrared spectroscopy), using a LactoScope C3 (Delta Instruments, PerkinElmer Company, Waltham, MA, USA): lactose, fat, protein, non-fat solids, and freezing point.

### 2.4. Determination of the Content of Heavy Metals and Trace Elements

Milk samples (100 mL for samples) were analyzed following the experimental protocol for determining the content of heavy metals in liquid samples with appropriate modifications [23]. After drying the milk on the stove, the chemical digestion of the resulting product was carried out. The ash was dissolved in 20 mL of 0.5 N HNO_3_ solution and filtered through ash-free filter paper before analysis. For each sample, the volume was brought to 50 mL with 30 mL of 0.5 N HNO_3_ solution. The nitric acid (65%, ρ = 1.39 g/cm^3^) used to prepare digestion solution (HNO_3_ 0.5 N) was purchased from Sigma-Aldrich Chemie GmbH (Buchs, Switzerland) and was trace metal grade (Supra-pur).

To assess Zn, Cd, Cu, Fe, Mn, and Pb concentrations in the filtrate, we used flame atomic absorption spectrophotometry with an acetylene–nitrous oxide flame (model ContrAA 300, Analytik Jena, Jena, Germany). Mixed standard solutions of heavy metals (1000 mg/L), namely, zinc (Zn), cadmium (Cd), copper (Cu), iron (Fe), manganese (Mn), and lead (Pb) (ICP Multielement Standard solution IV CertiPUR), were procured from Merck. For each analyzed TE, stock solutions (1000 ± 5 mg·kg^−1^ d.w.) were purchased from the May and Baker Group and prepared in three different concentrations for constructing the corresponding calibration curves. All glassware was treated with 20% (*v*/*v*) Pierce solution, rinsed with cold tap water, treated with 20% (*v*/*v*) nitric acid, and then rinsed again with double-distilled water. All blanks and duplicate samples were analyzed during the procedure. NCS Certified Reference Materials DC 85104a and 85105a (China National Analysis Center for Iron and Steel) were separately analyzed for quality assurance. The percentage recoveries for TE analysis varied between 85% and 105%. The percentage recovery means were as follows: Zn (102%), Cd (105%), Cu (105%), Fe (92%), Mn (95%), and Pb (94%). The variation coefficients were below 10%. Detection limits (μg/g) were determined using the calibration curve method: Zn (0.43), Cd (0.01), Cu (0.13), Fe (0.15), Mn (0.19), and Pb (0.05). The TE levels in waters were expressed as milligrams per kilogram dry weight (mg·kg^−1^ d.w.). All measurements were performed by the same researcher in the same conditions for all milk sampling.

### 2.5. Determination of Microbiological Parameters

The microbiological parameters analyzed were the total number of aerobic mesophilic germs that developed at 30 °C (TMB), the total number of coliform bacteria (TCB), the number of *Staphylococcus aureus*, and the number of somatic cells (SCCs). The determination of microbiological parameters was carried out in accordance with specific methodology for the analysis of milk samples: for mesophilic bacteria—ISO 4833-1:2013 [24]; for coliform bacteria—ISO 21528-2:2017 [25]; for *Staphylococcus aureus*—EN ISO 6888-1:1999 [26]. The somatic cell count (SCC) was determined using the Ekomilk Scan Somatic Cell Analyzer (Bulteh2000, Plovdiv, Bulgaria).

### 2.6. Statistical Analysis

Physicochemical data, chemical parameters, heavy metals, trace elements, and microbiological data from the milk samples were statistically analyzed using the non-parametric Mann–Whitney test. For variables with continuous datasets (numeric values) for all samples and timepoints, two-way ANOVAs were conducted, with treatment type and time used as factors (independent variables). The first factor included two groups: (i) without additional feed, and (ii) with additional feed; the second factor included three groups: (i) 31 days (1–31 May 2023), (ii) 61 days (1 May 2023–30 June 2023), and (iii) 92 days (1 May 2023–31 July 2023). Prior to statistical analysis, the measured values for these datasets were first log-transformed (decimal logarithmation) to reduce skewness. Datasets eligible for two-way ANOVA were tested for normality and homoscedasticity (homogeneity of variance) using Kolmogorov–Smirnov and Bartlett’s tests, respectively. For dependent variables meeting both these conditions, post hoc testing was conducted using the Neuman–Keuls methodology for each significant main effect. Comparisons for treatment type (as the main effect) were conducted against the treatment without additional feed (as the control); comparisons for time (as the main effect) were conducted against the earliest time point (as the control). For significant interactions between treatment type and exposure time, these comparisons were conducted at each timepoint using the treatments without additional feed as the reference groups. For the other dependent variables, paired *t*-tests were used to compare groups showing only continuous datasets. In this case, normality was checked as described above, homoscedasticity was tested using F tests, and inter-group differences were tested using *t*-tests. Differences were considered significant at *p* < 0.05.

## 3. Results

### 3.1. Physicochemical Parameters

The amount of milk obtained after milking was determined quantitatively; the results are presented as the mean ± standard deviation in Table 1. The values obtained for the amount of milk from the two experimental groups indicate significant increases in the amount of milk in the case of sheep from the second experimental group, i.e., those that benefited from additional feed.

Among the physical characteristics of milk, we considered the pH and the freezing point; the determined values are presented as the mean ± standard deviation in Table 2. The chemical parameters of the analyzed sheep milk were lactose, fat, protein, and non-fat solids (Snf), as also presented in Table 2.

### 3.2. The Content of Trace Elements and Heavy Metals

The assessment of milk quality was also based on the content of trace elements and heavy metals. Minerals in milk are essential for the human body; hence, determining possible contamination with heavy metals is necessary because it can lead to the alteration of human health. Trace elements and heavy metals are essential parameters to be determined from milk samples; the values determined for Zn, Fe, Cu, Pb, Cd, and Mn are shown in Table 3.

Datasets for zinc, iron, and manganese content in milk included only continuous (numerical) values for all groups; thus, they were analyzed using two-way ANOVAs. The corresponding values were normally distributed (*p* ≥ 0.158) and homoscedastic (*p* ≥ 0.065). The assumptions underlying the application of two-way ANOVA were, therefore, fulfilled for both factors (treatment type, treatment duration). A significant effect of treatment type on Zn levels was identified (F(1, 17) = 7.84, *p* = 0.016, η^2^ = 0.285). Similar effects were observed for milk Fe content (F(1, 17) = 8.85, *p* = 0.011, η^2^ = 0.331) and milk Mn content (F(1, 17) = 5.71, *p* = 0.034, η^2^ = 0.245). The effect of treatment duration was, however, statistically non-significant for all the aforementioned variables (*p* > 0.068). Post hoc analyses using the Neuman–Keuls procedure showed significant increases in Zn levels (*p* = 0.016), Fe levels (*p* = 0.011), and Mn levels (*p* = 0.034) for animals receiving additional feed.

In the case of milk Cd content, the measured values were below the limit of detection; therefore, no statistical analysis was possible for this variable. Such values were also detected for both treatment groups at 31 days. The measured values for milk Cu content were higher at 92 days than those assessed at 61 days (*p* < 0.001) for the animals not receiving additional feed. In contrast, no difference in Cu levels was observed at these two timepoints in specimens given supplementary feed (*p* = 0.628). Similar results were obtained for Pb concentrations in milk (*p* > 0.991).

### 3.3. Microbial Content

The analysis of milk from a microbiological point of view is essential because a series of diseases can be transmitted through milk; furthermore, these analyses also allow us to assess the health condition of the animals, taking into account milking hygiene rules and their degree of maintenance at a farm level. The values recorded for the microbiological parameters, i.e., somatic cell count (SCC), total mesophilic bacteria (TMB), total coliform bacteria (TCB), and *S. aureus*, are rendered in Table 4.

Datasets for SCC, TMB, TCB, and *S. aureus* contained only continuous (numerical) values for all groups. As a result, these datasets were analyzed using two-way ANOVAs. Since the measured values were normally distributed (*p* ≥ 0.098) and homoscedastic (*p* ≥ 0.234), the requirements needed for the application of the two-way ANOVA were fulfilled for both factors (treatment type, treatment duration). No significant effect of treatment type was observed for any of the aforementioned parameters (*p* > 0.432). The effect of treatment duration was also statistically non-significant (*p* > 0.147).

## 4. Discussion

The amount of milk produced by sheep from non-dairy breeds is usually lower than that produced by sheep from breeds specialized for milk production. The values determined in our study for mixed-breed sheep are in accordance: for group A, the average milk produced was 674.55 ± 5.50 g; for group B, the average milk produced was 745.55 ± 3.76 g. Administration of the feed complex determined a higher milk production in sheep from experimental group B, compared to the milk production obtained in animals where the classical feeding system was applied (*p* < 0.05). The amount of milk produced by the sheep in our study showed variations depending on both the lactation period and the feeding system. The values recorded herein are much higher compared to those obtained from milk cheeses [27]. Regarding the comparison of the amount of milk obtained from non-dairy breeds with the data from our study, the average DMY for the entire lactation period in the case of sheep from both groups was higher [8].

Furthermore, comparing our values with those obtained by other researchers who applied different feeding systems, similar values can be found regarding the increase in milk production depending on the applied feeding system [16,28]. Variations in milk production are influenced not only by breed, but also by other factors such as the feeding system, climatic conditions, and reproductive management.

The pH values for raw sheep’s milk vary between 6.4 and 6.8; these values are closely related to the chemical composition of the milk and the state of health of the sheep (mammary gland disorders). It is important to maintain the pH values within normal parameters because they affect the milk coagulation process and the manufacture of milk products. The pH values in our study mostly fell within the mentioned range. Compared to other pH values presented in other studies, the values recorded herein were similar to those determined in sheep specialized for milk production [13,29], as well as in non-dairy sheep [8].

The fat and protein content of the analyzed milk shows variations depending on the lactation period and the type of feed applied. Data from our study indicated that milk fat levels were highest in July (8.32%) and lowest in May (7.86%) and June (7.75%) for group A. In the case of milk from ewes in group B, the fat content showed higher values compared to those in the milk from group A (*p* < 0.05); the highest values were recorded in July (8.65%), followed by May (8.70%) and June (8.12%). The casein content in the milk samples from our study had values that recorded increases from May to July, with the values obtained in the milk samples from group B being higher compared to those from group A (*p* < 0.05). The fat content decreased in the middle of the lactation period, with a slight increase being observed toward the end. These variations were probably due to the change in composition of the meadows and the metabolic adaptations of the sheep to changes in the climatic, physiological, and nutritional conditions specific to the milking period [1].

The period of lactation influences the fat and casein content of the analyzed milk samples; this factor had an influence on both milk components, causing a gradual increase in their content. The results obtained in our study are in accordance with the data reported by other researchers [8,13,27,30]. The values obtained in our study were also compared with those from other studies that analyzed milk from sheep breeds specialized in milk production. We found that both the fat and the protein content were higher than or comparable to the reported data [1,27,29,31]. Moreover, comparing the values obtained for the fat and protein content of milk from sheep of the Turcana breed with those obtained from sheep of non-milk breeds, the recorded values were higher [8].

The application of a combined feeding system (pasture and complex feed) determined significant increases in fat and casein content in the analyzed milk samples, and these results were in agreement with those obtained in other studies [17,18,19].

The lactose content in the milk samples from both groups was balanced throughout the study, with values between 4.03% and 4.60% for milk from the sheep in group A, and values between 4.29% and 4.81% for milk from the sheep in group B. The lactation period did not have a significant effect on this parameter, with the small increases in the values not being significant from a statistical point of view. The results obtained are in agreement with those obtained by other groups of researchers who followed the effect of this variable on milk production [8]. The feeding system influenced the lactose content of the milk samples analyzed in our study, with the increases recorded in the milk samples from sheep in group B being significant compared to those from group A (*p* < 0.05). The results obtained in our study on the level of lactose in milk from sheep that received mixed feed are in agreement with those from other studies that analyzed the effects of different exploitation systems. The results suggest that, if the animals that received a mixed diet (grazing supplemented with the administration of a complex fodder), the composition of sheep milk was improved [17,18,19].

The feeding system influences the amount of milk, the chemical composition of milk, and its physicochemical characteristics. The content of fats, proteins, and lactose in the analyzed milk samples recorded significant increases for the sheep from group B, i.e., those that benefited from a mixed diet—grazing supplemented with the administration of a complex fodder prepared on the farm. The results obtained are in accordance with those from another study, which posited that the fatty acid composition and nutritional value of milk would be improved as a result of the introduction of grazing to the sheep’s diet [5].

Somatic cell count (SCC) represents one of the most important parameters allowing the assessment of the quality of sheep’s milk. SCC can be considered a very sensitive marker for assessing the health of the animal, mainly the health of the udder [32]. At the level of European Union legislation, Regulation 853/2004 sets a limit for the number of somatic cells in cow’s milk (400,000 cells/mL), but there is no limit for the number of somatic cells in sheep’s milk [33]. The USA has established a maximum SCC limit of 750 × 10^3^ cells/mL for sheep’s milk [34]. In contrast, in Romania, there is no legal limit for this parameter, and the threshold or limit of SCC in sheep’s milk is still a debated topic.

Previous studies carried out have allowed the researchers to establish thresholds for this parameter. Thus, to ensure a satisfactory hygienic and technological quality of sheep’s milk, a threshold of 70,000 cells/mL (6.85 log_10_ cells/mL) was proposed [35], and a somatic cell count of less than 500,000 (5.70 log_10_ cells/mL) was suggested for good-quality sheep [36].

In the case of our study, the values recorded in the milk samples from sheep in group A were 4.28–4.89 log_10_ cells/mL, and those from sheep in group B were 4.48–4.97 log_10_ cells/mL. Comparing the values obtained in our study with those presented in other studies, we can find that the values were higher than 2.05–2.04 log_10_ cells/mL [16] and 3.271–3.288 log_10_ cells/mL [5], comparable to 4.89 log_10_ cells/mL [37], and lower than 5.98–5.95 log_10_ cells/mL [38]. The content of SCC in milk samples is directly influenced by the observance and application of milking hygiene rules, breed, and other factors.

Another study suggested that the threshold for SCC in a healthy sheep should not be higher than 250,000 cells/mL (5.39 log_10_ cells/mL) [8]. The values obtained in our study are relatively low and balanced, and the lactation period and the feeding system did not have a significant effect on the SCC values in the analyzed milk samples, remaining above the threshold agreed by many groups of researchers.

Milk is synthesized at the level of the mammary gland; thus, its state of health influences the quality of milk from a microbiological point of view. However, milk can be colonized by a series of microorganisms during and after milking. Contamination of milk during milking and handling can occur at the surface of the nipple, feed, milking equipment, air, and water, as well as other environmental sources. Therefore, the microbiological quality of sheep’s milk is influenced by several factors: milking method, breed, housing conditions, season, stage of lactation, and degree of hygiene at the farm level [39].

According to Regulation (EC) 853/2004 of the European Parliament and the Council of the European Union [33], the number of mesophilic bacteria growing at 30 °C (TMB) is an essential indicator of the microbiological quality of sheep’s milk. Recommended TMB values for raw sheep’s milk to be pasteurized should not exceed 1.5 × 10^6^ CFU/mL (6.18 log_10_ CFU/mL). However, if raw sheep’s milk is intended for processing without heat treatment, its TMB may be at most 5.0 × 10^5^ CFU/mL (5.70 log_10_ CFU/mL) [33]. In the case of milk samples from sheep in group A, the values recorded for TMB were between 2.4 and 3.2 log_10_ CFU/mL; for the milk from sheep in group B, the values were between 2.1 and 3.2 log_10_ CFU/mL. The values obtained in the present study are lower compared to those provided in European legislation, as well as those obtained in other studies that applied a manual milking system [1,40].

At the level of raw milk samples, microorganisms can be identified that are capable of producing food spoilage; in the case of insufficient thermal processing of milk, these microorganisms can affect the health of consumers. The consumption of raw sheep’s milk can present a high risk for human health due to the presence of potential bacterial pathogens, including coliform bacteria and *Staphylococcus aureus* [41,42]. The values recorded for total coliform bacteria (TCB) in the case of milk samples from sheep in group A were between 1.3 and 2.3 log_10_ CFU/mL, compared to values were between 1.4 and 2.3 log_10_ CFU/mL from sheep in group B. *S. aureus* was identified in the sheep milk samples from the two groups; the recorded values were 0.5 and 1.8 log_10_ CFU/mL for the milk from sheep in group A, and 0.6 and 2 log_10_ CFU/mL for the milk from sheep in group B. No statistically significant variations were identified for any of the two bacteriological parameters.

Determining the content of trace elements and heavy metals in milk samples is important to prevent the risk of animal contamination, especially human contamination, as a result of the location of farms in polluted areas (mining areas, industrial areas, etc.). Milk can be used as a bioindicator of environmental quality, since the consumption of water or feed contaminated with heavy metals will lead to these elements being present in milk samples.

Zinc is an essential element involved in many physiological processes of the animal and human body. The amount of Zn in sheep’s milk and dairy products of ovine origin differs depending on the breed, the pedoclimatic conditions, the geographical location of the farms, the conditions and the systems of growth and feeding, and the plant composition of the grazing areas. The values presented in the literature for Zn vary in sheep milk samples, with values of 4.68 ppm [43], 4350 mg·kg^−1^ [44], 10.35 mg·kg^−1^, 25.65 mg·kg^−1^ [45], and 2683.5 ppb [10] being presented. In the case of our study, the amount of Zn recorded values between 13.68 and 15.78 mg·kg^−1^ for milk from sheep in group A, compared to values between 14.83 and 15.95 mg·kg^−1^ for milk from sheep in group B. Analyzing the average monthly values of Zn content in the milk samples from the two groups, we found significantly higher values in the case of the milk samples from sheep in group B (*p* < 0.05). The additional administration of feed complex containing micro- and macro-elements led to an increase in the value of the Zn concentration in the milk samples from sheep taken in the study.

Fe is a trace element that ensures a constant maintenance of the level of hemoglobin in the human body and, thus, prevents the onset of iron-deficiency anemia. The constant maintenance of Fe concentration in the body is achieved by consuming foods rich in iron, as well as by consuming milk and milk derivatives. The iron concentrations determined in the milk samples were between 1.32 and 3.72 mg·kg^−1^ for sheep in group A, and between 3.67 and 4.13 mg·kg^−1^ for sheep in group B. Analyzing the average monthly values of Zn content in milk samples from the two groups, we found significantly higher values in the case of milk samples from sheep in group B (*p* < 0.05). The additional administration of feed complex containing barley, maize, and micro- and macro-elements led to an increase in Zn concentration in the analyzed milk samples.

Cu is a trace element necessary for the proper development of cardiac and pulmonary activity, neuro-endocrine function, and iron metabolism; however, in the case of Cu poisoning as a result of consuming contaminated food, it can lead to poisoning and even death [10]. The level of Cu in milk and dairy products is carefully monitored; values in the literature vary widely, e.g., 0.011–0.498 mg·kg^−1^ [46], 0.34 mg·kg^−1^ [45], 0.02 mg/L [47], or values below the detection limit [45]. The results of our study are in agreement with those in the literature, ranging from 0.215 mg·kg^−1^ to the absence of Cu, where values were below the instrument’s detection limit (˂LOD). There were no significant differences in the samples of milk from sheep in the two groups, with the limits allowed in the legislation not being exceeded. Cu is a good indicator of environmental quality; its presence in high concentrations in soil, plants, and food products indicates an existing source of pollution.

Pb is a pollutant for food products; the presence of Pb in milk and dairy products is determined by environmental sources (atmosphere, vehicles, urban waste, mining operations, etc.). Pb is toxic for the animal and human body, and it has negative effects on human health; for this reason, maximum values allowed for milk samples and dairy products have been introduced. For example, the Food Code of Turkey provides values of 0.020 mg·kg^−1^ for Pb, but no limit is specified for other metals [10]. The level of Pb in milk samples varies widely in the literature, e.g., 0.06 ppm [40], 0.019–0.126 mg·kg^−1^ [46], ˂LOD, and 0.06 mg·kg^−1^ [45]. In our study, the recorded values for Pb levels were between ˂LOD and 0.07 mg·kg^−1^. No values were identified that exceeded the allowed limit, and no significant differences were found between the two groups. The presence of Pb levels below the detection limit (˂LOD) indicates the absence of pollution in the area where the microfarm was located.

Cd is an important food contaminant, which exhibits high toxicity and negative effects on human health as a result of the consumption of contaminated food. The contamination of milk and dairy products can stem from the environment (soil, fertilizer, atmosphere). The values of Cd concentrations in milk vary depending on the area of origin of the milk and the location of the farms, being in the vicinity or not of sources of pollution. Few studies have recorded the values of Cd in analyzed milk samples. Values of 0.63 ppm were reported in [43]; however, most studies reported Cd levels below the detection limit in analyzed milk samples [45,47]. The results of our study with respect to the level of Cd in the analyzed milk samples coincide with the results presented in the literature, being below the limit of detection (˂LOD). This finding indicates that there are no toxicological risks related to Cd pollution or Cd contamination in the western area of Romania, i.e., the location of the microfarm.

Mn is a trace element that acts as an enzymatic cofactor in the animal and human body, and it is necessary for the proper functioning of the body, especially the nervous system. Mn is provided through nutrition; for example, milk and dairy products contain Mn in their composition, but the content differs depending on the lactation period, the animal’s feeding system, and technological processes used to obtain milk products. The studies carried out on the content of Mn in milk samples from different regions have reported values of 0.010–0.179 mg·kg^−1^ [46], 0.076 mg·kg^−1^ [44], 0.01–0.26 mg·kg^−1^ [45], and 30.8 ppb [10]. In the case of our study, values of Mn between 0.195 and 0.268 mg·kg^−1^ for milk from sheep in group A, and between 0.196 and 0.279 mg·kg^−1^ for milk from sheep in group B were recorded. Analyzing the average monthly values of the Mn content in the milk samples from the two groups, we found significantly higher values in the case of milk samples from sheep in group B (*p* < 0.05). The additional administration of feed complex containing barley, as well as micro- and macro-elements, in its composition led to an increase in Mn concentration in milk samples from sheep of the Turcana breed.

Because heavy metals cause serious acute and chronic health problems, national and international food organizations have introduced regulations to prevent contamination. The legislation in force establishes the maximum limits allowed for these heavy metals in food products. Monitoring the content of trace elements and heavy metals in milk samples and dairy products is a European recommendation with which all farmers must comply.

## 5. Conclusions

The administration of the fodder complex produced within the microfarm determined higher values at the level of the analyzed milk samples for the following parameters: milk production, fat, protein, and lactose content, and the content of trace elements Zn, Fe, and Mn. The microbiological parameters analyzed at the level of all milk samples indicated a generally good state of health of the sheep, especially of the mammary gland, thus highlighting the absence of microbiological contamination of the milk, as well asa high degree of hygiene on the farm and in the milking process. On the basis of the values recorded for the concentrations of heavy metals, we can conclude that there are no sources of pollution in the area where the microfarm is located. The results obtained in our study, despite the relatively low number of evaluated sheep, suggest that sheep of the Turcana breed can also be used for milk production, provided that additional feed is administered. The high content of fats, proteins, and Zn, Fe, and Mn, along with the optimal pH and SCC values, determined in the milk samples from sheep benefiting from a mixed feeding system, resulted in sheep milk products of good quality. The results of this study indicate the need to continue studies on the identification of effective feeding strategies (e.g., the administration of fodder complexes) that will determine an increase in the quantity and quality of milk (with respect to the content of fatty acids with beneficial effects on human health) in sheep of the Turcana breed, while reducing economic costs.

## Figures and Tables

**Table 1 animals-13-02977-t001:** Daily milk yield from Turcana sheep.

Experimental Group/Daily Milk Yield (g)	Month			Total
	May	June	July	
**Group A (n = 100)**	710.66 ± 12.50	693.33 ± 8.50	619.66 ± 25.32	674.55 ± 15.44
**Group B (n = 100)**	775 ± 9.16 *	750 ± 24.02 *	711.66 ± 3.78 *	745.55 ± 12.32 *

* Indicates significant differences according to the Mann-Whitney test between the values of the 2 experimental groups (*p* < 0.05).

**Table 2 animals-13-02977-t002:** Physical characteristics and chemical composition of milk from Turcana sheep.

Parameters	Month	Group A (n = 100)	Total	Months	Group B (n = 100)	Total
**pH**	May	6.72 ± 0.035	6.74 ± 0.027	May	6.87 ± 0.090	6.81 ± 0.093
	June	6.81 ± 0.023		June	6.81 ± 0.091	
	July	6.69 ± 0.031		July	6.76 ± 0.100	
**Freezing point (°C)**	May	0.56 ± 0.003	0.57 ± 0.002	May	0.56 ± 0.003	0.56 ± 0.007
	June	0.57 ± 0.002		June	0.57 ± 0.013	
	July	0.56 ± 0.003		July	0.56 ± 0.005	
**Lactose**	May	4.03 ± 0.182	4.28 ± 0.167	May	4.29 ± 0.051	4.60 ± 0.132 *
	June	4.60 ± 0.036		June	4.81 ± 0.125	
	July	4.23 ± 0.291		July	4.70 ± 0.221	
**Fat**	May	7.86 ± 0.025	7.98 ± 0.030	May	8.7 ± 0.286	8.49 ± 0.246 *
	June	7.75 ± 0.030		June	8.12 ± 0.170	
	July	8.32 ± 0.036		July	8.65 ± 0.287	
**Casein**	May	4.68 ± 0.036	4.77 ± 0.047	May	5.2 ± 0.381	5.41 ± 0.282 *
	June	4.79 ± 0.070		June	5.30 ± 0.264	
	July	4.84 ± 0.036		July	5.74 ± 0.201	
**Non-fat solids (Snf)**	May	11.67 ± 0.041	11.52 ± 0.040	May	11.58 ± 0.040	11.53 ± 0.041
	June	11.45 ± 0.036		June	11.53 ± 0.046	
	July	11.45 ± 0.045		July	11.49 ± 0.038	

* Indicates significant differences according to the Mann-Whitney test between the values of the 2 experimental groups (*p* < 0.05).

**Table 3 animals-13-02977-t003:** Concentration of trace elements and heavy metals in milk from Turcana sheep.

Parameters	Month	Group A (n = 100)	Months	Group B (n = 100)
**Zn**	May	14.19 ± 0.477	May	15.14 ± 0.429 *
	June	14.97 ± 0.567	June	15.37 ± 0.463 *
	July	15.07 ± 0.900	July	15.87 ± 0.087 *
**Fe**	May	2.66 ± 1.163	May	3.76 ± 0.107 *
	June	3.33 ± 0.193	June	3.80 ± 0.156 *
	July	3.54 ± 0.187	July	4.06 ± 0.082 *
**Cu**	May	˂LOD	May	˂LOD
	June	0.17 ± 0.029	June	0.19 ± 0.010
	July	0.20 ± 0.007	July	0.19 ± 0.021
**Pb**	May	˂LOD	May	˂LOD
	June	0.06 ± 0.005	June	0.06 ± 0.005
	July	0.06 ± 0.005	July	0.073 ± 0.005
**Cd**	May	˂LOD	May	˂LOD
	June	˂LOD	June	˂LOD
	July	˂LOD	July	˂LOD
**Mn**	May	0.20 ± 0.011	May	0.21 ± 0.020
	June	0.21 ± 0.009	June	0.25 ± 0.039 *
	July	0.22 ± 0.036	July	0.26 ± 0.017 *

* Indicates significant differences according to the Mann-Whitney test between the values of the 2 experimental groups (*p* < 0.05).

**Table 4 animals-13-02977-t004:** Somatic cell count (SCC) and microbial counts in milk from Turcana sheep.

Parameters (log_10_ Cells/mL)	Month	Group A (n = 100)	Total	Months	Group B (n = 100)	Total
**Somatic cell count (SCC)**	May	4.62 ± 0.313	4.71 ± 0.154	May	4.69 ± 0.185	4.76 ± 0.152
	June	4.70 ± 0.113		June	4.84 ± 0.149	
	July	4.8 ± 0.036		July	4.75 ± 0.127	
**Total mesophilic bacteria (TMB)**	May	2.8 ± 0.400	2.86 ± 0.328	May	2.43 ± 0.351	2.74 ± 0.270
	June	2.86 ± 0.208		June	2.73 ± 0.305	
	July	2.93 ± 0.378		July	3.06 ± 0.152	
**Total coliform bacteria (TCB)**	May	1.90 ± 0.529	1.98 ± 0.362	May	1.80 ± 0.360	2.00 ± 0.220
	June	1.96 ± 0.351		June	2.00 ± 0.201	
	July	2.10 ± 0.210		July	2.20 ± 0.100	
** *S. aureus* **	May	1.33 ± 0.503	1.27 ± 0.427	May	1.36 ± 0.665	1.45 ± 0.528
	June	1.03 ± 0.472		June	1.33 ± 0.503	
	July	1.46 ± 0.305		July	1.66 ± 0.416	

## Data Availability

Not applicable.

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
