# Peer review of "Effects of Feeding Management System on Milk Production and Milk Quality from Sheep of the Turcana Breed"

_animals, 2023, doi:10.3390/ani13182977_

Round 1
Author Response
Thank you very much for the review and for the suggestions that are clearly meant to improve the quality of our manuscript. In what follows, we answer the requirements and hope that we have understood them correctly.
A1 – completed as suggestion - L55, L601-602
- Pulina, G., Nudda, A., Battacone, G., Cannas, A. Effects of nutrition on the contents of fat, protein, somatic cells, aromatic compounds, and undesirable substances in sheep milk. Anim. Feed Sci. Technol. 2006, 131, 255-291. doi:10.1016/j.anifeedsci.2006.05.023.
A2 – completed as suggestion – L59-60
A3 – 19. Sanz Sampelayo, M.R., Chilliard, Y., Schmidely, P., Boza, J. Influence of type of diet on the fat constitutes of goat and sheep milk. Small Rumin. 2007, 68(1), 42-63. DOI: 10.1016/j.smallrumres.2006.09.017. – L618-619
A4 – completed as suggestion – L90, L618-619
A5 – completed in the 2.2 Animal feeding and management the components of ProteMix. –L146-149
A6 - We agree with what was mentioned in the comment. In our study, only the 2 mentioned factors were taken into account. Thank you for the suggestion, we will certainly take the suggestion into account in the following studies. – L321
A7 - completed as suggestion –L384-387
A8 - completed as suggestion - 34. Kaskous, S., Farschtschi, S., Pfaffl, M.W. Physiological Aspects of Milk Somatic Cell Count in Small Ruminants-A Review. Dairy. 2023, 4, 26-42. https://doi.org/10.3390/ dairy4010002 – L393-394, L656-657
A9 - The text has been reformulated according as suggestion – L545-548
A10 - The text has been reformulated according as suggestion. – L550-554

Reviewer 2 Report
I present my considerations about the scientific article, making the descriptions as below.
along the lines of the document, namely:
Lines 33, 179, 182, 391 and 578: put scientific names in italics.
Line 116: The authors should insert a table containing the chemical composition of the concentrated feed. No less important is to present what product was used, since there are 3 products in this brand (ProteMix by HL-TopMix), all of then prepared for feeding cattle. I believe the authors used ProteMix® Lactating Cows. The authors should justify why they used a product prepared for feeding cattle. Thus, the authors must present the minimum levels of nutrients in the feed guaranteed by the manufacturer, the ingredients in the feed and possible substitutes.
Line 123: What were the preparation procedures for milking? Was any preservative substance used in the sterile containers?
Line 163: put -1 in superscript and put the dot between mg and kg as follows: (1000±5 mg.kg-1 d.w.)
Line 174: put the dot between mg and kg
Table 2: Freezing point in oC and not in C
Line 283: non-dairy breeds is different from non-lactating breeds. The correct term is non-dairy breeds.
Line 279: The amount of milk produced by animals in group B was, in fact, different from group A. However, for the amount of concentrated feed used, the increase in milk production was small. Even so, the authors did not consider the possible variation in the animals' body weight and body condition score, even though these characteristics were not considered as parameters in the present study. The number of animals used (N) in each of the treatments would allow an interesting evaluation of these parameters. Why were they not measured?
Line 336: animals are supplemented with nutrients. It does not exist. Diets are supplemented. Supplementing means adding what is potentially deficient in the diet, that is supplying those nutrients that may be limiting the performance of the animals. In the line 341 above, the authors writted: a mixed diet – grazing supplemented with...
Lines 338 to 341: Definitely, there is no information in the present study that supports this type of statement (The content of fats, proteins and lactose in the analyzed milk samples recorded significant increases in the milk samples derived from oil from group B). By the way, the presentation of nutrient levels assured by the feed manufacturer was requested. The greater supply of non-fiber carbohydrates in the rumen from the concentrated feed (maize and barley) contributed to the greater supply of nutrients to the host animal, especially short-chain fatty acids (SCFA) and amino acids (ProteMix® Lactating Cows). From the metabolism of amino acids and propionate, it was possible to increase the formation of glucose by gluconeogenesis in the liver. This glucose is indispensable for the formation of lactose, which causes water to enter the alveolar lumen of the mammary gland, thus increasing the volume of milk. The increased amount of milk fat may be partially due to the greater availability of acetate in the mammary gland.
Line 428: wheat? It was not presented in the Materials and Methods.
Line 554: doi is not correct. Put 2 in the end of DOI: https://doi.org/10.5194/aab-65-407-2022
Author Response
Thank you very much for the review and for the suggestions that are clearly meant to improve the quality of our manuscript. In what follows, we answer the requirements and hope that we have understood them correctly.
I present my considerations about the scientific article, making the descriptions as below. along the lines of the document, namely:
Lines 33, 179, 182, 391 and 578: put scientific names in italics. - corrected as suggestion
Line 116: The authors should insert a table containing the chemical composition of the concentrated feed. No less important is to present what product was used, since there are 3 products in this brand (ProteMix by HL-TopMix), all of then prepared for feeding cattle. I believe the authors used ProteMix® Lactating Cows. The authors should justify why they used a product prepared for feeding cattle. Thus, the authors must present the minimum levels of nutrients in the feed guaranteed by the manufacturer, the ingredients in the feed and possible substitutes. – completed as suggestion the chemical composition of ProteMix. L146-149
Line 123: What were the preparation procedures for milking? Was any preservative substance used in the sterile containers? - The milk was milked in sterile containers, without the addition of preservatives.
Line 163: put -1 in superscript and put the dot between mg and kg as follows: (1000±5 mg.kg-1 d.w.) - corrected as suggestion. L198,
Line 174: put the dot between mg and kg - corrected as suggestion. L209
Table 2: Freezing point in oC and not in C - corrected as suggestion.
Line 283: non-dairy breeds is different from non-lactating breeds. The correct term is non-dairy breeds. – corrected as suggestion. L-324
Line 279: The amount of milk produced by animals in group B was, in fact, different from group A. However, for the amount of concentrated feed used, the increase in milk production was small. Even so, the authors did not consider the possible variation in the animals' body weight and body condition score, even though these characteristics were not considered as parameters in the present study. The number of animals used (N) in each of the treatments would allow an interesting evaluation of these parameters. Why were they not measured? - Thanks for the suggestion, yes - in the present study the possible variation of the body weight and the body condition score of the animals were not taken into account. We think the results would have been interesting. In the following studies, we will definitely analyze the 2 parameters, respectively the way they influence the quantity and quality of the milk produced.
Line 336: animals are supplemented with nutrients. It does not exist. Diets are supplemented. Supplementing means adding what is potentially deficient in the diet, that is supplying those nutrients that may be limiting the performance of the animals. In the line 341 above, the authors written: a mixed diet – grazing supplemented with... - corrected as suggestion L377-378
Lines 338 to 341: Definitely, there is no information in the present study that supports this type of statement (The content of fats, proteins and lactose in the analyzed milk samples recorded significant increases in the milk samples derived from oil from group B). By the way, the presentation of nutrient levels assured by the feed manufacturer was requested. The greater supply of non-fiber carbohydrates in the rumen from the concentrated feed (maize and barley) contributed to the greater supply of nutrients to the host animal, especially short-chain fatty acids (SCFA) and amino acids (ProteMix® Lactating Cows). From the metabolism of amino acids and propionate, it was possible to increase the formation of glucose by gluconeogenesis in the liver. This glucose is indispensable for the formation of lactose, which causes water to enter the alveolar lumen of the mammary gland, thus increasing the volume of milk. The increased amount of milk fat may be partially due to the greater availability of acetate in the mammary gland. - The composition of ProteMix has been completed in the Materials and methods chapter. From the composition of the product, it results that a surplus of protein, cellulose and fats is ensured. L146-149
Line 428: wheat? It was not presented in the Materials and Methods. – corrected as suggestion L475
Line 554: doi is not correct. –
Put 2 in the end of DOI: https://doi.org/10.5194/aab-65-407-2022 - completed as suggestion L611

Reviewer 3 Report
Throughout the text separate words with infen appear incorrectly, for example L36, L138, etc...
Confirm what the acronym FTIR means
The presentation of the units should always be the same throughout the text. Sometimes it appears (mg kg-1), other times (mg/kg) it should even out the formatting of the units.
Staphylococcus aureus and other Latin names should be written in italics.
L185 - Italicize the entire subtitle
Table 2 and 4 are unformatted, data centered others at the bottom
Author Response
Thank you very much for the review and for the suggestions that are clearly meant to improve the quality of our manuscript. In what follows, we answer the requirements and hope that we have understood them correctly.
Throughout the text separate words with infen appear incorrectly, for example L36, L138, etc... - Corrected as suggestion
Confirm what the acronym FTIR means – completed as suggested. L180
The presentation of the units should always be the same throughout the text. Sometimes it appears (mg kg-1), other times (mg/kg) it should even out the formatting of the units. Corrected as suggestion. L457-460, L481, L484, L493-497, L520-522
Staphylococcus aureus and other Latin names should be written in italics. - Corrected as suggestion. L33, L438
L185 - Italicize the entire subtitle - Corrected as suggestion
Table 2 and 4 are unformatted, data centered others at the bottom - Corrected as suggestion
